# Machine Unlearning Doesn't Do What You Think: Lessons for Generative AI Policy and Research

A. Feder Cooper[1,2,3*‡]    Christopher A. Choquette-Choo[4*]    Miranda Bogen[5,6*]

Kevin Klyman[3*]    Matthew Jagielski[4*]    Katja Filippova[4*]    Ken Ziyu Liu[3*]

Alexandra Chouldechova[2]    Jamie Hayes[4]    Yangsibo Huang[7]    Eleni Triantafillou[4]

Peter Kairouz[7]    Nicole Mitchell[7]    Niloofar Mireshghallah[8]    Abigail Z. Jacobs[9]

James Grimmelmann[1,10,11]    Vitaly Shmatikov[10]    Christopher De Sa[12]    Ilia Shumailov[4]

Andreas Terzis[4]    Solon Barocas[2]    Jennifer Wortman Vaughan[2]    Danah Boyd[2]

Yejin Choi[8]    Sanmi Koyejo[3]    Fernando Delgado[13]    Percy Liang[3]    Daniel E. Ho[3,14]

Pamela Samuelson[15]    Miles Brundage[16]    David Bau[17]    Seth Neel[18]    Hanna Wallach[2]

Amy B. Cyphert[19]    Mark A. Lemley[14]    Nicolas Papernot[4]    Katherine Lee[1,4*‡]

[1]The GenLaw Center    [2]Microsoft Research    [3]Stanford University    [4]Google DeepMind
[5]Center for Democracy & Technology    [6]Princeton    [7]Google    [8]University of Washington
[9]University of Michigan    [10]Cornell Tech    [11]Cornell Law School    [12]Cornell University
[13]Lighthouse    [14]Stanford Law School    [15]UC Berkeley    [16]Independent
[17]Northeastern University    [18]Harvard Business School    [19]W. Virginia University, College of Law

## Abstract

"Machine unlearning" is a popular proposed solution for mitigating the existence of content in an AI model that is problematic for legal or moral reasons, including privacy, copyright, safety, and more. For example, unlearning is often invoked as a solution for removing the effects of specific information from a generative-AI model's parameters, e.g., a particular individual's personal data or the inclusion of copyrighted content in the model's training data. Unlearning is also proposed as a way to prevent a model from generating targeted types of information in its outputs, e.g., generations that closely resemble a particular individual's data or reflect the concept of "Spiderman." Both of these goals—the targeted *removal* of information from a model and the targeted *suppression* of information from a model's outputs—present various technical and substantive challenges. We provide a framework for ML researchers and policymakers to think rigorously about these challenges, identifying several mismatches between the goals of unlearning and feasible implementations. These mismatches explain why unlearning is not a general-purpose solution for circumscribing generative-AI model behavior in service of broader positive impact.

## 1 Introduction

Since around 2016, technical experts and policymakers have invoked machine unlearning as a way to operationalize compliance with an individual's "right to be forgotten" in the E.U.'s General Data Protection Regulation (GDPR) [112], with respect to removing personal data from deployed models. More recently, with the emergence of generative AI, machine unlearning has captured public attention as a potential general-purpose approach for purging unwanted information from generative-AI models and systems. More and more, research papers, policy briefs, and media reports suggest that unlearning

---

*First author; ‡Project lead. Corresponding authors: {afedercooper, kate.lee168}@gmail.com

39th Conference on Neural Information Processing Systems (NeurIPS 2025) Position Paper Track.

can help meet a broad range of objectives for both open and closed models and systems,[1] spanning privacy, copyright, safety, and other issues [e.g., 9, 55, 70, 77, 85, 106, 127, 141, 147].

For generative AI, the goal of unlearning has shifted beyond the traditional ML focus on removing the influence of specific training data from model parameters. This kind of *targeted removal* is often infeasible in this setting, and even when possible, it may not prevent problematic content from appearing in a model's outputs.[2] As a result, unlearning now also includes *targeted suppression* at the output level, reflecting an attempt to manage what generative-AI models produce, not just what is encoded in their parameters. But these are fundamentally different goals—both in the technical methods they require (Section 3) and in how their outcomes may be treated under specific legal provisions (Section 5). As a result, mapping unlearning methods onto policy objectives is not straightforward; important mismatches and slippages arise, for three overarching reasons:

**Deleting information from an ML model is not like deleting data from a database.** There is no way to cleanly identify, target, and delete specific, contained pieces of information from an ML model's parameters. Instead, it is possible (though expensive) to train a *new* model on a dataset that does not contain problematic data (Section 3)—e.g., a specific scientific paper on designing novel viruses or a specific in-copyright image of Spiderman. This is typically what it means to "remove" data from a model in unlearning, which deviates from intuitive understandings of the term. Removal applies to *discrete pieces of data* in the training dataset *before training occurs*; it cannot target latent patterns a trained model has learned across different data—e.g., more general concepts of "how to synthesize a toxic molecule" or "Spiderman." There is also no obvious or appropriate way to go about translating such open-ended aims to concrete tasks that can be implemented by an algorithm (Section 4).

**Removing information from model *parameters* does not give guarantees about model *outputs*.** It is possible to retrain a model so that specific data are removed *from the training dataset*, and thus those removed data do not influence the model's parameters. But doing so will not necessarily prevent the model from producing outputs that resemble removed data. For instance, even if one removed all in-copyright images of Spiderman from a model's training data, this does not mean it would be impossible for that model to produce outputs that resemble Spiderman at generation time. Generative-AI models are impressive in part because they generalize beyond the information that is exactly contained in their training data. It is therefore a mistake to think that making a limited set of targeted changes to a model's parameters is sufficient to make promises or guarantees about what types of outputs that model could or could not possibly generate (Section 4).

**Suppression methods are necessarily imperfect.** While it is technically feasible to suppress certain types of outputs, suppression techniques (e.g., output filters) are unlikely to produce perfect legal compliance. This is in part because generative-AI systems exhibit familiar but fundamentally irresolvable tensions that are inherent to highly generative technologies [159], like the PC and the Internet (Section 6). Just as a PC could be used as a tool to perpetrate fraud or to write the next great Broadway musical, a generative-AI system can similarly be put to malicious or beneficial uses. Suppression methods cannot control downstream use; this would require anticipating how a person or other agent might behave with outputs in a potentially infinite number of contexts—none of which is reasonably under the purview of a technical method alone (Section 4).

We explore each of these points in detail. Altogether, we articulate fundamental, conceptual mismatches between technical methods in machine unlearning (Section 4) and aspirations for the broader impact that these methods could have for law and policy (Section 5 & 7).[3] While machine unlearning research will continue to progress, we are doubtful that unlearning methods will deliver full compliance in the future. Nevertheless, unlearning techniques may offer limited benefits that support certain outcomes for law and policy. In summary:

**Section 2.** We discuss evolving motivations for machine unlearning in generative AI—to extend beyond the *targeted removal* of the influence of specific training data on a model's parameters to also include the *targeted suppression* of specific content from model outputs.

**Section 3.** We briefly discuss common technical approaches for both removal and suppression.

**Section 4.** We show how our discussion illuminates five conceptual mismatches between unlearning motivations and concrete technical methods—mismatches that make clear there are substantive aims that cannot, from first principles, be addressed with unlearning methods alone.

**Section 5.** We examine how these mismatches manifest differently and exhibit various implications for U.S. copyright law. (We defer related analysis on privacy and safety to the Appendix.)

**Section 6.** In light of these limitations, we provide recommendations on how ML experts should focus their research and how policymakers can adjust their expectations and norms concerning reasonable best efforts when relying on an unlearning method in practice.

## 2 Background and motivations for machine unlearning

**Early motivations for supervised machine unlearning in the GDPR.** In some jurisdictions, individuals have rights associated with the control of their personal data. Notably, since its adoption in 2018, Article 17 of the E.U.'s GDPR provides the "Right to erasure" (more commonly called the "right to be forgotten") [48, 112], which gives individuals broad rights (with exceptions) to demand that companies delete their personal data.[4] ML researchers often interpret Article 17 to apply to both to the individual's data examples that have been used as training data and to the resulting trained models themselves [e.g., 10, 24, 83, 84, 93, 99, 115] (Appendix A).[5] This presents a problem because, in almost all cases, the model would not *just* be trained on a specific, right-exercising individual's data. It would also be trained on data associated with thousands of others, if not many more. Wholesale erasure of a trained model, in response to one individual's deletion request, would therefore likely be an extreme, over-broad interpretation of Article 17.[6]

This problem raises a natural question for ML research: rather than deleting a trained model altogether, is it possible to develop algorithms that can achieve more targeted removal of training data from the model? In the specific context of ML research's common interpretation of the GDPR: is it possible to remove the influence of the right-exercising individual's data from the model, without imposing on the model controller the undue burden of the cost of retraining a new model from scratch without that individual's data examples (Section 3.1)?

Machine unlearning is the area of ML research that attempts to address this question. In the technical literature, this area corresponds to a wide variety of different techniques, which are loosely grouped together. For this reason, we will rely on a loose, intuitive (rather than rigorous) definition of machine unlearning that captures this common underlying technical motivation: machine unlearning is a sub-area of machine learning that develops methods for the *targeted removal* of the effect of training data from the trained model. This definition encompasses work from the last 10 years that has studied unlearning in clustering [57], classification and regression [11, 45, 110, 131], federated learning [73, 94], and more [17, 154]. It also applies to the classic paper by Cauwenberghs and Poggio [19], which studies the problem of unlearning in support vector machines (SVMs) under the name "decremental learning" over two decades ago. Further, this definition is deliberately broad, rather than prescriptive. We intentionally do not include specific requirements for *how* certain information is "targeted" or "removed."[7] For now, we also are not prescriptive about what the exact "effects" are of "learned information" on the trained model's behavior. We will address this in more detail in the sections that follow.

**Evolving motivations for unlearning in generative AI.** Translating prior unlearning methods from supervised settings to generative AI exhibits some important technical challenges. Supervised ML tends to involve models that produce concise outputs from a bounded and typically fixed set (e.g., classifications like `dog` or `cat`). After using an unlearning method, a model's outputs for a given input may change (e.g., may flip from `cat` to `dog`), but the set of possible outputs generally remains the same. In contrast, for generative-AI models, the set of possible outputs is significantly more expansive. With this key difference, the desired goals for what machine unlearning could achieve have expanded— beyond *removal* of the influence of *training-data inputs* on the trained *model's parameters*—to also encompass desired effects on the model's possible *generated outputs* when the model is *put to use*.

That is, the loose definition for unlearning has recently widened in scope: machine unlearning aims to develop methods for (1) the *targeted removal* of the effect of training data from the trained model and (2) the *targeted suppression* of content in a generative-AI model's outputs. But these are two very different goals. They are different in terms of the technical methods they involve (Section 3), and their results might not be treated similarly with regard to specific provisions in law (Section 5). For instance, consider an individual's training-data deletion request to exercise their "right to be forgotten" under the GDPR. Such a deletion request might have nothing to do with *suppressing* a model's outputs to not reflect that individual's personal data. Nevertheless, research on unlearning for generative AI attempts to address both of these types of problems.[8]

It is an appealing idea that machine unlearning could serve both of these ends. If so, it would also perhaps be reasonable to assume, as many researchers and organizations have, that machine unlearning could on its own be used to solve issues related to problematic model outputs in a variety of policy-relevant domains: novel privacy challenges [12, 23, 83, 104, 156], copyright [26, 46, 87, 144, 158], safety [90, 91, 95], and more. However, as we discuss below (Section 4), thinking about removal and suppression interchangeably or as being in service of the same ends can lead to confusion about what unlearning methods can achieve for operationalizing compliance with legislation (Section 5).

## 3 Unlearning methods and evaluating evidence for their success

To get to the root of this confusion, we first need a bit more background on how technical methods for *removal* and *suppression* differ. Originating from work in supervised machine learning, removal methods aim to eliminate the influence of targeted training data on a model's parameters (Section 3.1). These methods typically act directly on the training data, before a model is trained. In contrast, suppression methods target information in a trained model's outputs (Section 3.2).[9] The training process instills models with complex patterns that are *latent* in the training data, and which can manifest as novel outputs when models generalize during inference or generation. Because suppression methods focus on outputs, they most often[10] target **latent information**.

### 3.1 Methods for removal (of information directly observed during training)

Unlike removing an entry from a database, there is no way to cleanly identify, target, and delete a specific training example from an ML model's parameters. This is because model parameters are not directly or easily interpretable. As a result, "removal" of information from a generative-AI model deviates from intuitive understandings of the term "removal." Instead, it is possible to remove problematic examples from the training data and to train a *new model* from scratch.[11] For instance, we can train a new text generation model without using data from a particular web domain. If there are no data from that web domain observed in the training process, then these data cannot have affected the model's parameters in the first place.

The approach of **retraining from scratch** is often referred to as the **"gold standard"** for machine unlearning [e.g., 59, 92, 100]. At first glance, this seems like a reasonable (albeit expensive) solution to the unlearning problem. However, the "gold standard" only directly targets information that is directly observable in the training data.[12] As a result, it may not be effective for ensuring unwanted information is not latent in the trained model's parameters, nor for preventing unwanted information from manifesting in the model's outputs at generation time (Sections 3.2 & 4).[13] In practice, implementing the "gold standard" is expensive—often prohibitively so for today's enormous models trained on enormous datasets by expending enormous computing resources.[14] This cost has motivated the development of more efficient methods for removal of *structured* information in the training dataset: methods that produce models that have similar properties to those that have been retrained from scratch.

Broadly speaking, there are two overarching approaches for removing structured information in the training data. First, methods for **structural removal** use custom procedures to reduce the amount of retraining that needs to be done to guarantee the exclusion of targeted training data. [e.g., 11, 152]. For this reason, structural removal is commonly referred to as **exact unlearning** in the ML literature [e.g., 150]. Even though these methods are different from the "gold standard," they retain the *exact* same guarantees of the "gold standard," with respect to removing the effect of targeted training data. (We avoid the term "exact unlearning" because it can be reasonably misunderstood to mean that such methods are able to "exactly" or "perfectly" unlearn anything; however, these methods do not apply to latent information that is encoded in a perhaps unidentifiable—i.e., unstructured—way in the model.[15])

Second, there are methods that approximate structural removal, often by changing the original model's parameters rather than retraining from scratch. These algorithms involve proofs (with specific theoretical assumptions) that the modified model is (by some mathematical definition) "similar" to a model that has been retrained from scratch [63, 84]. Of course, such approximations are not literally equivalent to retraining from scratch; they often involve a probabilistic guarantee, not absolute certainty, that the targeted information has been successfully removed. As such, these approximate methods are often referred to as **inexact unlearning**.[16] Further, nearly all of these methods have been developed for supervised ML, not generative AI, and they do not immediately translate to this new setting.[17]

## 3.2 Methods for output suppression

Most unlearning methods in generative AI focus on output suppression. Potentially problematic training data are observed during the training process, and there is no attempt to guarantee (with certainty or probabilistically) that this is not the case. Instead, these methods attempt to make the generation of undesirable content less likely—but they do not guarantee that the model could never produce such content. These methods tend to be more computationally feasible than retraining from scratch and they also apply (to varying degrees of success) to latent information.

There are two overarching approaches to output suppression: (1) methods that *modify the trained generative-AI model*, and (2) methods that leave the model unchanged, but *modify the system in which model is embedded*. While it is now common to include these methods under "machine unlearning," arguably, they have nothing to do with "unlearning" some information from a model; they bear more resemblance to **alignment** techniques. Methods that modify the model attempt to direct it away from being able to produce outputs that reflect undesirable content (e.g., through additional training or model editing) [72, 97, 100, 102, 105, 153, 157].[18] System-level interventions include guardrails like **output filters**, which are wrapped around the model to prevent generations that contain certain undesirable content from being surfaced to end users [137].[19] These filters may themselves be implemented with ML models (e.g., classifiers), which exhibit greater or lesser degrees of precision and accuracy.[20]

The success of output suppression methods is most often evaluated by examining how they affect the types of generations that are produced in some downstream task. This often involves prompting the model or system with respect to content that the method intended to suppress, and observing if the resulting generations do not reflect that information [e.g., 20, 46, 100].[21] For instance, in safety contexts, evaluations often rely on the WMDP benchmark [91], which is a multiple-choice question dataset that focuses on biological, chemical, and cyber-security risks. One might test the original model on this question dataset as a baseline, and then apply an output suppression method, re-test, and quantify changes in the answers as a proxy for determining if "unsafe" knowledge is no longer reflected in the model's answers [e.g., 134].[22]

## 4 Fundamental mismatches between unlearning motivations and methods

Five intertwined problems emerge directly from our discussion of removal (Section 3.1) and suppression (Section 3.2). Output suppression is not a replacement for removal of training data (Mismatch 1). Conversely, removal of training data does not guarantee meaningful output suppression (Mismatch 2). More generally, models are not equivalent to their outputs (Mismatch 3) or to how their outputs are put to use (Mismatch 4). And last, because *targeted* removal and suppression are challenging to implement, unlearning can have unintended consequences (Mismatch 5). We address each in turn.

**Mismatch 1** *Output suppression is not a replacement for removal of training data.*

With output suppression, it is possible that a particular piece of information is still represented in the model's parameters, and that this information could manifest in or impact the model's outputs (Mismatch 3).[23] These details could have important consequences for law and policy. If a piece of legislation were to call for the explicit removal of a piece of training data from a model's training dataset, unlearning methods that fall short of guaranteeing structural removal would likely not suffice [53]. In other cases, modifications to the model or system to suppress certain types of training data in outputs may be sufficient. In general, the appropriateness of unlearning methods for removal or suppression to operationalize compliance with legislation will depend on the exact details. These details include the particular legal domain in question. They may also include the circumstances of the use that exposes information that was meant to be addressed with unlearning (e.g., if some atypical, adversarial usage pattern is necessary for exposure of problematic information in the training data).

**Mismatch 2** *Removal of training data does not guarantee meaningful output suppression.*

Exactly removing a piece of data from a model's training dataset does not guarantee that it would be impossible for the model to generate outputs that resemble it. Consider deleting a particular phone number. Given latent information the trained model may contain about other phone numbers (and about numbers in general), it may be possible for the model to generate a specific phone number for which all associated training data have been removed (Appendix D.1). Similarly, one could remove

all in-copyright images of Spiderman from an image generation model's training dataset and retrain from scratch (Section 3.1). But again, this does not guarantee that the new model could not possibly produce an output that might be "substantially similar" to copyrighted expression of Spiderman, based on how the model generalizes from latent information derived from the information that remains in its training data (Section 5.1). In both cases, removal could perhaps make the generation of similar outputs less likely; however, this cannot be assured in general.

From both of these examples, our main point is that there is a meaningful slippage that occurs when employing a removal technique in service of the goal of output suppression: it is unclear which set of information should be targeted for removal from the training data in order to prevent the generation of certain outputs. Removal of a narrow set of information (e.g., training data that contain certain phone numbers) can easily be *under-inclusive*. Being *over-inclusive* is also a potential problem, especially for cases that attempt to handle indeterminate concepts like "Spiderman." One could remove all information related to comic books, spiders, the colors blue and red, the humanoid form, etc. But this is too broad: it may be effective at preventing generations that reflect "Spiderman," but it also removes significantly more information that one did not originally intend to target (Mismatch 5) [e.g., 71, 98].[24]

Both sides of these examples—over-inclusiveness and under-inclusiveness—clarify how the "gold standard" (Section 3.1) can be challenging to implement and interpret as a baseline for unlearning. Implementing the "gold standard" requires navigating difficult, if not arbitrary, trade-offs to draw boundaries around what exactly to include for removal. One could choose to retrain without all in-copyright images of Spiderman that they manage to identify in the training data, but this would not necessarily include pictures of people in Spiderman Halloween costumes (Section 5.1). How to make these choices is clearly not a straightforward task, and yet it is essential when evaluating a particular unlearning method against the "gold standard" as a baseline, in order to make judgments about its efficacy.

**Mismatch 3** *Models are not equivalent to their outputs.*

The slippage discussed above runs deep, and relates to a more general mismatch in unlearning research. Notably, it is typical to evaluate the success of an unlearning method not by examining changes in the model's *parameters*, but by prompting the model and measuring the extent to which certain types of *outputs* are no longer generated (Section 3.2). This has consequences for how we should think about gauging the success of an unlearning method. For instance, consider that an individual $p_0$ has associated data examples and that a model trainer retrains a model from scratch without those examples. But now consider that there are also training-data examples related to individuals $p_1, \ldots, p_n$ that are by some quantitative measure similar to $p_0$'s. A user prompts the retrained model with some (perhaps public) information about $p_0$ (e.g., demographics, address), with the goal of revealing information about $p_0$'s health status. Combining the latent information in the model (from training on data concerning $p_1, \ldots, p_n$) with the additional information provided in the user's prompt about $p_0$, the model generalizes to produce an output that reveals sensitive information about $p_0$'s health status. This problem is related to what Shumailov et al. [124] calls "ununlearning": "unlearned knowledge gets reintroduced in-context, effectively rendering the model capable of behaving as if it knows the forgotten knowledge."

**Mismatch 4** *Models are not equivalent to how their outputs are put to use.*

A corollary follows from Mismatch 3, which involves another slippage. Among those who mistakenly believe that unlearning is a standalone solution for effectively moderating possible model outputs, some reason further that this could help curtail further downstream undesirable or malicious model uses in practice.[25] It is obvious, but nevertheless important, to emphasize that seemingly innocuous outputs could be put to undesirable downstream uses. To greater or lesser extents, different unlearning methods can remove the effect of specific training data from models or suppress certain types of model outputs; but the type of control this provides is localized to the model. Additional control would require anticipating how a person or other agent might behave with model outputs in an unbounded number of contexts—none of which is reasonably under the purview of machine unlearning.

**Mismatch 5** *Unlearning can have unintended consequences.*

A final point, which we have alluded to above, is that various forms of unlearning may have unintended consequences. Even if a particular method is successful at removing a specific piece of information from a model or suppressing its appearance in outputs, it is often the case that the method will also remove or suppress *other* information that the implementer did not intend to target. For

example, removing a chosen set of facts from the training dataset might change how a model answers questions about seemingly unrelated facts. Similarly, output suppression is likely to affect not just outputs that include the material intended to be suppressed, but also other outputs. In both cases, the effects are not necessarily predictable, and may compromise model utility [11, 60, 75, 84, 92].[26]

# 5   Machine unlearning in policy and practice

We next consider how these five mismatches manifest in specific ways and introduce complications for U.S. copyright, where researchers and organizations have suggested unlearning could help resolve key issues introduced by generative AI [e.g., 2, 80, 81, 151]. Copyright is just one example. In Appendix D.1, we address analogous lessons for privacy, where we argue that unlearning is not an adequate remedy for data deletion requests and suppressing outputs that resemble personal information, as well as other technical challenges (e.g., identifying all instances of relevant training data to target within large-scale datasets). Similarly, in Appendix D.2, we analyze the mismatches we identify for safety considerations, arguing that there are unclear boundaries for what to target for removal or suppression, given that such decisions touch on inherent tensions in dual-use systems.

Each of these examples demonstrates in practice the fundamental, conceptual mismatches that we describe in Section 4, revealing an even deeper disconnect between the feasibility of using of unlearning methods, actual policy considerations, and regulatory compliance. To address this disconnect, judges and policymakers will need to set reasonable expectations concerning the imperfect outcomes of best-effort implementations of unlearning methods to support specific policy goals (Section 6).

## 5.1   U.S. copyright

At first glance, unlearning may seem like a desirable approach to mitigating copyright infringement in generative-AI models. Unlearning methods could target either exact duplication of copyrighted expression in model outputs, or copying of more general, latent information that relates to creative expression—perhaps as a way to operationalize notice-and-takedown requests [41].[27] However, U.S. copyright is not a straightforward problem, and unlearning is not a straightforward solution.[28]

Copyright law protects "original works of authorship fixed in any tangible medium of expression" [36]. This means that copyright protection extends to original (modestly creative) works, like a particular image or a particular paragraph of writing, but not to any ideas or facts contained in it. Because copyright law gives creators the exclusive right to prepare reproductions (copies) and derivative works, courts examine whether potential copies are "substantially similar" in expression to the original work, and thus whether those copies infringe on the rights of the copyright holder.[29] Substantial similarity is a challenging concept with a varied and complicated history in copyright caselaw. Common tests to determine substantial similarity are subjective [119]; judgments for substantial similarity cannot "be reduced to a simple formula that can easily be applied across different works and genres" [87, p. 72].

Building a training dataset, training a model, and producing substantially similar copies at generation time may all involve copies of copyrighted works. Not all such copying is copyright infringement. Depending on the circumstances, the defense of "fair use" may protect uses of a work for a different purpose (potentially including training a model) and some uses that modify the work in "transformative" ways. Both "fair use" and "transformative use" are technical terms in copyright law [37, 89]. Whether a particular use is fair depends on the facts of each case, including the effect on the market for the copyrighted work in question. In the most analogous situations, courts have held intermediate copies made for purposes of generating new outputs to be fair use [6, 88]. Such intermediate copies may include copies of training data, which are reorganized into batches to serve as inputs for model training. However, courts are less likely to hold the output of a generative-AI model or system is fair use if it is substantially similar to the original—unless it parodies or comments on the original [67, 87]. This may include regurgitated training data [18, 35, 109], or outputs that are substantially similar to those training data.

Following from this brief background, we focus our discussion of copyright and machine unlearning on training data and generated outputs. We do not address potential implications for intermediate artifacts, e.g., a trained model's parameters [26, 35].

**Suppression of substantially similar outputs.**   If a model generates an output that is substantially similar to a copyrighted work, in response, it may be tempting to try to use unlearning methods

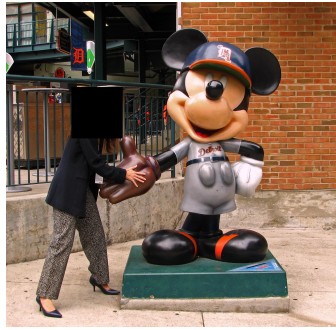
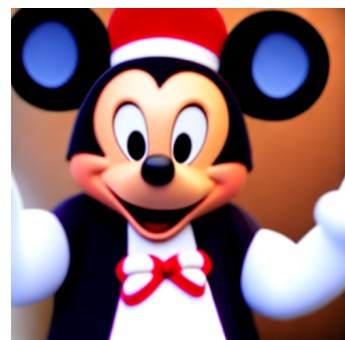

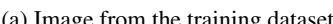

(a) Image from the training dataset      (b) Generation for the prompt `"Mickey Mouse"`

Figure 1: One can think of CommonCanvas [60] as a "gold-standard" model that does not contain in-copyright images of Mickey Mouse: the only training data that contain Mickey Mouse expression are from personal photographs, e.g., (**a**). Even without unlicensed, in-copyright training images of Mickey Mouse, the model can generate outputs that resemble "Mickey Mouse," e.g., (**b**).

to remove the ability to do so. For the reasons discussed above concerning suppression methods (Section 3.2), this is challenging because there is no notion of similarity that can be used to programmatically and comprehensively determine which works are substantially similar in the interest of copyright law [87, 119].[30] A feature, like a color scheme, may be problematic if copied from one work but not from another work (so long as it is copyrightable subject matter). The techniques used to suppress generations similar to a particular in-copyright image of Mickey Mouse may not generalize to suppressing generations that are similar to another image of Mickey Mouse—or of any other Disney character [87, Part II.F]. In general, while such techniques may limit problematic outputs in some cases, they cannot generalize to all cases.

Further, in order to suppress certain outputs, for example, those that resemble in-copyright expression of "Spiderman," the overall generative-AI system likely needs to have learned information about this expression in order to *not* present it to the end user. An output filter would need to be able to identify "Spiderman" (likely, from being trained on data that contain "Spiderman"-related expression) in order to filter it out. So, even if one were to remove all instances of "Spiderman" from the generative-AI *model*, more generally, it might be infeasible to remove all information about "Spiderman" from the generative-AI *system*. Moreover, suppression at a sufficient level of generality would likely also suppress other noninfringing content, including both fair uses of Spiderman in parody, and depictions of other superheroes with similar color schemes (Mismatch 5).

**Removal of specific training examples.** Removing a particular training example from the model's training dataset, by contrast, is a narrower approach that is less likely to be overreaching (Section 3.1). But it may still overreach, since copyright does not forbid all forms of copying (e.g., fair uses, internal copies [62]). Removal of an example could prevent transformative, non-infringing uses of the example in addition to potentially infringing ones (Mismatch 2). None of the unlearning methods we have described can or do distinguish between transformative fair uses and non-transformative superseding uses,[31] and transformativeness is not the only relevant factor for fair use. It is unreasonable to expect unlearning methods to capture these nuances. As evident from caselaw, courts themselves struggle to draw the line of fair use.[32]

Removal may also be ineffective in preventing infringing outputs. Even with removing a set of relevant data examples, it may still be possible to generate an output that is similar to the removed data (Mismatch 2). In such cases, this can be due to the presence of elements of the original work in other works in the training data [35, 78], for example, related works, duplicates, or otherwise similar works that themselves may or may not be deemed infringing copies of the original work. For a concrete example, consider CommonCanvas, a text-to-image generation model, for which the training dataset's images all have Creative Commons licenses [60]. The training dataset does not contain reproductions of unlicensed, in-copyright images of Mickey Mouse; and yet, based on inclusion in the training dataset of licensed personal photographs (e.g., from Disney World), it is still possible for Common-Canvas to generate images that could be judged substantially similar to "Mickey Mouse" (Figure 1).[33]

Finally, because copyright is not limited to exact duplication, efforts to have a model unlearn copyrighted material are likely to be significantly overbroad. The problem is not just that some uses

of the exact material, like criticism or parody, are permitted. It is that in order to prevent a model from generating a cartoon that looks too similar to Mickey Mouse, we might have to make it unlearn other concepts that are not themselves infringing, like other cartoon mice or rodents.

**Unlearning methods as tools for causation.** If a generated output is alleged to be too similar to a particular plaintiff's creative work, the plaintiff (e.g., an artist) will have to prove copying to establish copyright infringement.[34] Defendants (e.g., a generative-AI company) may attempt to use a counterfactual argument to challenge causation. That is, if a model had been trained without the inclusion of the plaintiff's work, would the model still produce output substantially similar to the plaintiff's? If so, that may seem to suggest that the presence of the plaintiff's particular work in the training dataset for the original model did not cause the output. This is significant in copyright law because independent creation [50] is a defense to infringement [123]. In this case, the "gold standard" of retraining the model from scratch could be used to produce a counterfactual model without the plaintiff's work.

This reasoning is tempting but incorrect. Whether this counterfactual model was trained without the plaintiff's work is remarkably hard to assure [35, 78]. There may be other copies or derivative works of the plaintiff's work in the training data (Figure 1). Those derivative works may have reasonable fair use arguments, but their existence in the training data may still influence the output. The converse, if a plaintiff can show that the output would have been significantly different without the plaintiff's work, is perhaps more convincing, but also flawed. This is because training is non-deterministic.[35] Two models *trained on the same dataset* (let alone different ones) may generate significantly different sets of outputs for the same prompt.[36] Judging whether these sets of outputs are meaningfully (and perhaps subtly) different is not a straightforward task to evaluate—neither with technical tools in machine learning [29, 143] nor with respect to making judgments about similarity for copyright [26].

In all, these difficulties show that, while unlearning techniques may seem appealing for copyright remedies, judges and practitioners must be careful to consider their current capabilities and limitations. Suppression methods could be deemed acceptable if courts accept the empirical evaluations of these methods; but judges who consider unlearning as a remedy to copyright infringement will have to weigh the practical limitations of these methods, as well as the potential unexpected consequences of unlearning on unrelated content. This is particularly relevant because copyright law imposes significant penalties for noncompliance, including statutory damages [40], destruction of infringing artifacts [39], and even criminal sanctions [38].

## 6 Takeaways for ML research and AI policy

Following from our discussion, we offer five takeaways for ML researchers and AI policymakers.

**Unlearning is just one approach in the ML and policy toolkit.** There are clear gaps for what unlearning can do to achieve policy aims, both with respect to removal and output suppression. Different methods may be useful to certain degrees in specific contexts, but it is important to view unlearning as just one approach among many others (e.g., acceptable use policies and responsible AI licenses [82, 101]) that could sometimes help achieve specific policy aims. Nevertheless, ML researchers should not claim—and policymakers should not misunderstand—that machine unlearning alone is generally effective for making generative-AI models and their outputs compliant with any desired policy goals.

**Evaluation of an unlearning method for a specific domain is a specific task.** General claims about the broader impacts of unlearning are likely to be wrong from first principles because each legal and policy regime has its own unique features, which can be subtle and nuanced. To make rigorous claims, as much as possible, ML experts need to evaluate specific techniques against specific regimes. This requires an understanding of these specific regimes, not just generalized ideas of how they might work—generalized ideas that may be so oversimplified that they are misleading or incorrect. For example, to make claims about the relevance of an unlearning method for U.S. copyright compliance, it is important to understand the nuances of what the law does and does not forbid [87].

The appropriateness of a particular technical mitigation hinges on these specifics. They cannot be overlooked or abstracted away. As we have seen throughout, these specifics can illuminate whether removal or suppression is the right technical goal to pursue for a specific substantive end. Further, in considering these specifics, it becomes unclear if removal of information from a model is the most relevant technical end to pursue for law and policy impact. In many cases, it seems like output suppression is what interested parties really care about (Sections 2), and so output suppression is perhaps

a more relevant area of focus for ML research that aspires to influence policy. A clear understanding of the specific goals of specific pieces of law or policy is also important for guiding the right set of solutions—technical or otherwise. In some regimes, perfect guarantees may be an unnecessary or undue burden for model developers and custodians. Reasonable efforts to remove or suppress particular information may be sufficient in some legal contexts, even if their results are imperfect. Of course, this will depend on the needs judges and policymakers articulate for the particular domain.

**Understanding unlearning as a generative-AI *systems* problem.** Systems-level interventions (e.g., content filters) are an important tool for constraining outputs (Section 3.2); evaluating such interventions clearly requires systems-level analysis [28, 113]. Open-weight models, like Meta's Llama models [96], therefore present different challenges for unlearning [35]. On their own, these models cannot implement system-level guardrails, with the intention of unlearning or satisfying any other purpose. In general, in order to achieve this type of functionality, developers who use open-weight models would need to implement mechanisms for output suppression in their own systems.

**Setting reasonable goals and expectations for unlearning.** It is also important for judges and policymakers to realize that, in general, it is unlikely that technical solutions for unlearning will improve significantly anytime soon. Instead, it will be important for policymakers to adjust their expectations for machine unlearning. This necessarily includes thinking through specific policy goals and, when using technical methods to achieve those goals, what should constitute reasonable best efforts in different contexts with respect to removal or suppression. For example, judges or regulators may expect best efforts to result in specific training data being removed from a model and related information suppressed from its outputs; and, if a company meets this bar, they would not seek massive fines if the model's outputs still somehow approximate that information. We expect the focus would then be on the remediation process—i.e., did a developer take reasonable steps—and not perfect results.

**There are no general-purpose solutions to constrain generative technologies.** Finally, and more generally, policymakers should resist the tendency to think unlearning can lead to generative-AI models that can do "everything but X." One of the strongest appeals of many generative-AI systems is that they are general-purpose. They can be adapted to a wide range of uses and produce a wide range of useful outputs. A superficial understanding of machine unlearning is that it can surgically and completely remove specific capabilities from a model while leaving everything else about the model unchanged. As we have seen, this is not what unlearning methods actually accomplish. The same power to abstract and generalize that makes these models so useful also means that, with small targeted changes, they are often still capable of exhibiting similar behavior.

This lesson is familiar from other generative technologies like the PC and the Internet [32, 159]. Ed Felten calls it the "Fallacy of the Almost-General-Purpose Computer" [52]. For example, the PC has the ability to be adapted to a wide range of computational tasks through suitable configuration and inputs; this means that there is no simple or reliable way to prevent a computer (let alone a generative-AI system) from ever being used to violate privacy, infringe copyright, or design a dangerous molecule—not without fundamentally compromising the flexibility and power that make it so useful. A toaster cannot design a bioweapon, but a toaster also cannot do much besides make toast. A computer can do many things, and so can a generative-AI system. This is an inherent tension with all generative systems. We can try to tether or constrain different aspects of this generativity in different ways, but we will not be able to block its capacity for harmful uses with one, comprehensive method—from either technology or policy—that will work in all possible contexts.

# 7 Conclusion

Machine unlearning involves different approaches for removal and suppression (Sections 2 & 3), which have very different consequences and side effects. Our goal has been to connect the technical and conceptual gaps at the core of these methods (Section 4) with policy expertise (Section 5), to help scientists and policymakers make sense of the choices, limitations, and costs of using unlearning methods for compliance (Section 6). Although we focus on copyright as a central example, our insights apply more broadly. In any legal or policy context—not just privacy and safety, which we explore in the Appendix—we can ask how the core conceptual mismatches involving unlearning play out in practice. This perspective can help clarify whether removal or suppression is the right goal to pursue in a given setting, and the costs each approach would impose.

## Acknowledgments and Disclosure of Funding

We thank the individual co-organizers of Evaluating Generative AI Systems: the Good, the Bad, and the Hype (GenLaw DC) for early conversations about machine unlearning that helped spark this collaboration. We also thank The K&L Gates Initiative in Ethics and Computational Technologies at CMU, The GenLaw Center, Georgetown Institute for Technology Law & Policy, and the Center for Democracy & Technology, who co-hosted the GenLaw DC workshop. We thank the reviewers and attendees of the *2nd Workshop on Generative AI + Law* at *ICML '24* for their feedback on earlier versions of this work. Lastly, we thank Jared Bomberg, Doug Burger, Nicholas Carlini, Susan Dumais, Alexandra Givens, Milad Nasr, Adam Roberts, and Jon Small for useful discussion and feedback on this piece.

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

# Notes

1 Our observations address cross-cutting issues applicable to both open and closed generative-AI technology—and technologies that lie on the spectrum between these two poles.

2 One important area where removal may be necessary (but still not sufficient) concerns illegal or otherwise prohibited content that may have been included in the model's training data, such as child sexual abuse material (CSAM) [111]. See Section 5.

3 Our contributions require expertise in ML, law, and policy. We intend for our audience to be members of all of those communities. We organized a team of experts in each discipline. The resulting paper reflects the efforts of a large-scale collaboration across academic institutions, civil society, and industry labs. We intend for this paper to be a standalone document: one with the necessary (and sometimes elementary) background information to make our contributions legible to our diverse intended audience, and at the appropriate level of abstraction to encourage effective cross-disciplinary communication about machine unlearning.

4 There are several exceptions to this right, for instance, when keeping the data is in the public interest (e.g., the data are used to comply with a legal ruling) or for certain research purposes, etc. [112, Article 17(3)].

5 While this interpretation is common in unlearning papers, it is still very much up for debate. Exactly how Article 17 may or may not apply to ML models is under active discussion. (See Appendix D.1.)

6 Also consider that many thousands of people might make such a request, each one requiring retraining a new model from scratch (Section 3.1). Retraining runs could perhaps be periodically batched—removing multiple individuals' personal data together in the same run to reduce the number of times a model is retrained. Still, retraining could happen a large number of times for the same model.

7 There is arguably a spectrum between *targeted* machine unlearning and unintentional (and undesired) loss of representation of information in the model, which prior work has studied in various settings, for example, **catastrophic forgetting** [61, 125]. Our focus is the former. See also Section 4, Mismatch 2.

8 In this respect, goals for machine unlearning research are similar to goals in other ML subareas that study privacy, such as differential privacy.

9 We provide additional discussion of information targets for machine unlearning in Appendix C. Our treatment of specific unlearning methods for removal and suppression is fairly brief. We deliberately do not provide an in-depth survey or taxonomy of state-of-the-art techniques that are branded as machine unlearning methods. Several groups of authors have already done so from different perspectives [e.g., 92, 118, 126]. Instead, our purpose here is to provide sufficient framing to elicit important conceptual gaps and limitations—fundamental mismatches between unlearning motivations, targets, and methods (Section 4). It is these mismatches that are the heart of our paper, and are relevant for understanding misalignment with law and policy aims (Section 5).

10 For useful models, most learning is generalization; however, models also memorize (near) exactly a portion of their training data [e.g., 18, 26, 35, 51, 65, 87, 109, 117, 130]. Understanding the relationship between memorization and generalization is an active area of research. Output suppression methods may also target memorization of observed information (Appendix C).

11 This is in some—though, as we will see, limited—senses an easier technical problem to solve. Even though we cannot treat a model like a database, we can treat a training dataset like one: observed information (Appendix C) can be relatively easily and directly identified, targeted, and removed from the dataset *before training* transforms this information and makes its effects difficult to locate in the model's parameters.

12 It may have an indirect effect on latent information in the model's parameters, but, given that the relationship between observed and latent information is not completely understood (Appendix C), this effect cannot be guaranteed.

13 This is why we put the term "gold standard" in quotation marks, which are typically absent in the technical literature. These limitations also have broader implications, which we examine in Section 4, Mismatch 2.

14 This is particularly so because, to be effective, retraining from scratch requires discarding all prior training efforts and any benefits of learning in existing models that included the data to be removed.

15 Further, structural removal methods do not guarantee effective output suppression. See Section 4, Mismatch 2.

16 Practical implementations do not always align with theoretical mathematical assumptions. In such settings, methods may still work reasonably well empirically, but they may lose their respective (exact or inexact) theoretical guarantees.

17 Most methods for (approximate) structural removal have been developed for traditional ML settings, not generative AI. There are a few methods for generative-AI contexts that have drawn inspiration from this work [e.g., 22, 83]. However, for two overarching reasons, traditional AI methods do not naturally translate to this newer setting. First, since structural removal methods typically require specific training processes for the original model, they cannot be applied to trained models that did not use those processes. This means that existing models that were not trained with structural removal in mind, such as Llama 3 405B [96], cannot post hoc be made compatible with these methods. Second, both structural removal methods and methods that approximate them are very computationally expensive at generative-AI scale [92]. For both of these reasons, removal algorithms are challenging to implement for generative AI in practice. Later, we will discuss how these practical challenges have important implications for legislative requirements around data deletion for production generative-AI systems (Appendix D.1).

18 They all use back-end modifications to the trained model to try to alter the model's outputs at generation time on the front end. (See Appendix B for more on this terminology and distinction.) As we have noted throughout, this is challenging to do in a *targeted* way because the relationship between model parameters and model outputs is not straightforward or, in some cases, possible to determine (Appendix C & Section 3.1).

19 Similarly, a system developer could implement **input filters** that filter problematic user prompts [113]—e.g., a filter that flags the user's prompt to generate the chemical formula for smallpox, and prevents the prompt from ever being supplied as an input to the model.

20 Other proposed methods utilize the **system prompt** for output suppression. A system prompt is a piece of developer-chosen text that the system adds internally to the context of all user-supplied prompts, often to coax the model away from producing generations that contain undesirable content [115, 137]. Such in-context mechanisms may or may not work in practice; they are generally imprecise.

21 There are various other types of evaluations, for example, probing latent information in the model. We defer to Lynch et al. [98] for further discussion on evaluation strategies.

22 In practice, given the open-ended "information rich" outputs of generative-AI models, it is very challenging (and an open research area) to come up with methods that reliably measure properties of model and system outputs [143]. Evaluation benchmarks like WMDP attempt to mitigate this complexity by setting up tasks (in this case, multiple-choice questions) that constrain the open-endedness of generated outputs.

23 As in our discussion of shifting goals for machine unlearning (Section 2) and concrete unlearning methods (Section 3), we continue to see a slippage between what model *is* (i.e., what is stored in its parameters) and the outputs that a model *could produce*. We attend to this in more detail in Mismatch 3.

24 It is in a sense possible to make unlearning effective by being so over-inclusive—by removing or suppressing so much information—that the model loses its ability to produce *anything* useful. However, it is also arguable that this is not a successful application of unlearning, since it is not "targeted" in a meaningful way.

25 Similar observations have been made in algorithmic fairness contexts: a model that produces risk scores for criminal recidivism is distinct from the distribution of scores that model produces over a given population, which is again distinct from how the (distribution of) scores gets used for decision-making, e.g., a magistrate using those risk scores to inform their judgments about whether or not to grant a defendant bail upon rearrest [e.g., 7, 34]. Nevertheless, this slippage takes on an expanded meaning for generative-AI contexts.

26 This is why some research evaluates if the application of an unlearning method has effects on information that was *not* intentionally targeted—i.e., if metrics for overall model utility are preserved [11, 75, 84, 92].

27 See Lee et al. [87, Part II.G] for more detailed discussion on Section 512 and generative AI.

28 We limit our specific discussion to U.S. copyright. Other jurisdictions exhibit differences in copyright doctrine and caselaw, for example, with respect to exceptions to copyright-holders' exclusive rights. While we draw from U.S. doctrine and caselaw, the overarching points that we make in this section about unlearning, substantial similarity, and causation have relevance to other jurisdictions.

29 For infringement, it is not sufficient for a second work to be based on an earlier one; there must still be substantial similarity in some expression. Courts have thus far rejected claims that all outputs of generative-AI models are derivative works and that the models themselves are derivative works. See Cooper et al. [35] for additional discussion.

30 *Rentmeester* [119] illustrates that two works can be substantially similar in a general way and not infringe copyright; infringement requires for the works to be substantially similar in their expressions.

31 A **superseding use** is when, in the market for an original work, a new work replaces an original work (e.g., purchases of a fourth edition of a textbook replace purchases of the third edition). A non-transformative superseding use is a superseding use in which the new, replacing work does not change the character or purpose of the original work (e.g., a freely circulated digital PDF of a textbook dramatically changes the market for for-purchase hard copies of the textbook) [16]. Having a transformative purpose (in the sense that the second work has a different purpose than the original) is also relevant for fair use.

32 This challenge extends beyond unlearning methods. Distinguishing between transformative fair uses and non-transformative superseding uses requires context (e.g., how will the generation be used?) that is typically not currently available in generative-AI systems.

33 Here, the training process still has **access** to images that contain information related to "Mickey Mouse," even if those images are not exact copies of unlicensed, in-copyright images of Mickey Mouse. Access to a copyrighted work is one type of evidence for proving copying in a copyright infringement suit [5]. See also [87, Part II.D].

34 Further, even if some outputs are substantially similar to a protected work and could constitute *prima facie* infringement, they may not qualify as infringing reproductions or derivative works—especially if they are never distributed or publicly used. For example, if someone generates an image of Mickey Mouse but keeps it private, it is unlikely they will be held liable for infringement.

35 One source of non-determinism is randomness in the order in which examples are surfaced to the model during training. Example ordering has an effect on the ultimate trained model's parameters and the models outputs [31, 44, 66].

36 Even a single model may do so, with different settings of sampling hyperparameters. This challenge is not unique to unlearning or copyright. For example, it presents challenges for successful membership inference attacks (MIAs) [66]—a common approach in ML to determine if a particular piece of data was a part of a model's training dataset.

## A   Growth of unlearning papers over time

We provide some cursory evidence to support that there has been massive growth of machine unlearning research in the last few years (Figure 2). Since we draw our results from arXiv, they do not include mentions of machine unlearning in technical reports (e.g., Bengio et al. [9]) or other literature outside of computer science (e.g., Floridi [53]).

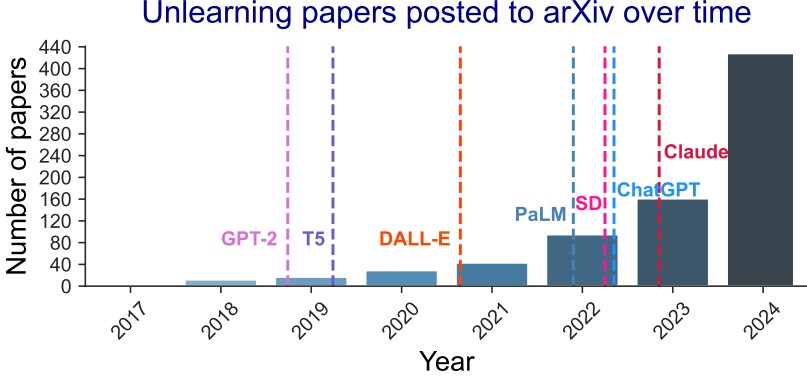

Figure 2: We scrape all papers that match `unlearn*` or `model forgetting` from arXiv and plot their counts over time, as of December 4, 2024. As of this date, there were a total of 810 papers starting from 1997 that matched out query. We indicate some important dates in the release of contemporary language and image generation models: **GPT-2**, **T5**, **DALL-E**, **PaLM**, **Stable Diffusion (SD)**, **ChatGPT**, and **Claude**.

GDPR was passed in 2016, and went into a effect in 2018. 790 of the papers have posting dates starting in 2016 (i.e., only 20 papers precede 2016). Of these 790 papers, 106 (i.e., 13.1%) mention

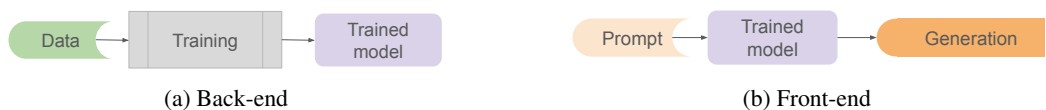

|                  |                  |
| :--------------: | :--------------: |
| (a) Back-end     | (b) Front-end    |

Figure 3: Both the (**a**) back-end and (**b**) front-end involve processes that have their own inputs and produce their own outputs (simplified here). This is why we use this additional terminology for clarifying which inputs and outputs are under discussion. There is nothing complicated here; it is just shorthand to signal different aspects of the trained model at different points in time.

"GDPR," "the right to be forgotten," or "RTBF" in the abstract. (These 106 papers are all from 2020-2024.) Given that we do not search the contents of all of the papers for these phrases, this serves as a lower bound of machine unlearning papers that reference GDPR.

We also manually coded each paper with different categories, which we then used to assist with our literature review for the paper. Note that, as of December 4, there have been more unlearning papers (428) posted to arXiv in 2024 than there were in all prior years combined. While not easily visible in Figure 2, there were 3 papers in 2017.

## B    Additional notes on evolving motivations for unlearning

As discussed in Section 2, machine unlearning originated as an area of supervised machine learning research that concerned the targeted removal of the effect of specific training data from trained models, such as classifiers. For generative AI, the scope of unlearning has expanded significantly: beyond *removal* of the influence of *training-data inputs* on the trained *model's parameters*—to also encompass desired effects on the model's possible *generated outputs* when the model is *put to use*.

In other words, the scope for unlearning no longer just concerns what Cooper and Grimmelmann [26] refer to as **back-end** considerations: "characteristics and capabilities of the *model itself* that directly result from its training." Unlearning also concerns **front-end** considerations: "how the *model behaves* in generating outputs in response to ... specific prompt[s]"(emphasis added) [26]. Both the back-end and front-end involve processes that have their own inputs and produce their own outputs (Figure 3). On the back-end, the training dataset is an input and the trained model is an output; the back-end involves making choices for which training data to include, which training algorithm to run, etc. On the front-end, the prompt is an input and the generation is an output; the front-end includes the processes of **inference** and producing generations, system-level filters that may prevent the processing of certain undesirable user prompts or the user-facing output of certain undesirable generations (Section 3.2), etc. On the back-end, the trained model *is an output*; on the front-end, the trained model *is used to produce outputs*.

This back-end/front-end terminology can serve as useful shorthand for distinguishing the different points in time where unlearning is of interest, and which artifacts a particular unlearning method is intended to affect—the *model parameters* or the *model's possible generations* (Section 3). Using the words "input" and "output" can sometimes be unclear, as they are overloaded with different meanings at different stages. (Also note that this is a different usage of "back-end" and "front-end" from Internet software, where "back-end" refers to server-side components like storage and "front-end" refers to client-side components like a user interface.)

## C    Targets for Machine Unlearning

Our loose definition for unlearning is abstract (Section 2); in fact, it is so abstract that it allows for an enormous number of reasonable interpretations and possible techniques that satisfy it. In this appendix, our aim is to provide some language that can help us be more precise. Building on our loose definition, we pin down useful ways to think about what types of information one might want to target with unlearning. In the main paper, we informally reference these targets; they are places where concrete unlearning methods could potentially apply in practice (Section 3). When we discuss mismatches (Section 4), we also note that some articulated goals for unlearning escape these target definitions altogether, highlighting instances where unlearning methods, from first principles, could not be applied rigorously or reliably for certain desired ends.

We define three overarching (and, as we will see, overlapping) **targets**: **observed information** (Definition 1), **latent information** (Definition 2), and **higher-order concepts** (Definition 3). These definitions build upon each other and become more abstract and indeterminate. This indeterminacy surfaces challenges for designing and implementing unlearning methods (Sections 3 & 4).

**Definition 1 Observed information.** *Data that are explicitly presented to the model during training. These data serve as inputs to computations that update the model's parameters.*

Observed information includes training-data examples. For instance, consider that the text "Susan's phone number is 555-123-4567" is included as an example in an LLM's training data. Since this text is used directly to train the LLM, it is observed information. Observed information also captures sets of training examples, such as all examples in the overall training dataset that mention Susan. It also includes data contained within examples, such as just the phone number "555-123-4567" in "Susan's number is 555-123-4567."

Effective trained models **generalize**: the learning process instills models with complex patterns that are derived from the observed information in the training data—patterns that models can apply to previously unseen information when they are put to use for inference or generation. This learned information is latent in the training data. Removal methods target observed information directly (Section 3.1).

**Definition 2 Latent information.** *Data that are not explicitly presented to the model during training, but that can be derived or otherwise elicited from a trained model based on the patterns that the model has learned during training.*

Unlike observed information (Definition 1), latent information is not *literally* observed in the training data. (However, there are ML-based methods that claim to identify latent information and make it observable in the trained model's parameters [e.g., 8, 43, 56] or indirectly through a model's outputs when the model is put to use [e.g., 25, 64, 107, 108].)

Latent information can include simple deductions [74, 121]. For example, given the observed information "Carlos is going to Susan's house for a birthday party this Thursday" and "Susan lives in Philadelphia," a possible piece of latent information is that Carlos is going to be in Philadelphia on Thursday. (Of course, just as with observed information, there is no guarantee that latent information is factually correct. In this example, perhaps Carlos attends the party remotely over a video call.) This information is not literally contained in the training data; it is derived from relationships learned from observed information.

Latent information can also be significantly more complex than such simple deductions. The power of large-scale models trained on enormous datasets [86] comes from their flexibility to capture all sorts of latent information—across observed information, across latent information, or across some combination of the two. Indeed, information can interact to produce sophisticated, higher-order information that ML research often refers to as "knowledge" or "capabilities."

**Definition 3 Higher-order concepts.** *Combinations of latent and observed information that manifest in the model as complex and coherent abstractions, knowledge, capabilities, or skills.*

Before giving some examples of higher-order concepts, some disclaimers are in order. Definition 3 is not intended to suggest something particularly deep about how models organize information or exhibit complex behaviors. (This is not, after all, a paper about ontology or metaphysics.) Instead, we give a definition of higher-order concepts for convenience: to align with how the ML technical literature tends to refer to conceptual learned representations. But it is nevertheless reasonable to think of higher-order concepts as complex combinations of latent information—that there is, loosely speaking, a spectrum of complexity for latent information (Definition 2), with simple deductions drawn directly from observed information on one end, and significantly more complex patterns (often called capabilities or emergent abilities [122, 132, 135, 145]) at the other.

This spectrum reveals that Definition 3 is somewhat arbitrary, since it is not clear how to distinguish when a piece of latent information is sufficiently complex to be considered a higher-order concept. We do not attempt to draw these lines. Nevertheless, we still find it useful to have a target definition that lets us to refer to the unlearning of higher-order concepts, since this is a type of information that could reasonably be—and, in some cases, is claimed to be—a target for machine unlearning (Sections 3, 4 & 5).

Given these disclaimers, we enumerate a few examples that satisfy Definition 3. A model's representation of a "person" [116] is a higher-order concept. "People" is also a higher-order concept (perhaps generalized from latent information about relationships between different "person"s). So, too, is the knowledge that composes concrete subjects like "Spiderman," "Marie Curie," and "basketball;" knowledge of abstract ideas like "justice" and "toxicity;" and notions of "artistic style" and "scientific phenomenon" (as well as instances of particular artistic styles and phenomena, like "Cubism" and "gravity"); and the ability to reason about the relationships between different concepts, including "mathematical reasoning."

Latent information and higher-order concepts are targets of suppression methods (Section 3.2). Some work on unlearning claims that removal methods target specific observed information in order to mitigate the effects of related latent information, but there are inherent conceptual (as well as feasibility) challenges that come about from attempting to do so (Sections 3 & 4).

# D    Additional sections on AI policy

In the main text, we provide detailed analysis on unlearning and U.S. copyright (Section 5). In the longer version of our paper [33]), we also include analysis for privacy (Appendix D.1) and safety (Appendix D.2)—two other policy areas where machine unlearning is frequently suggested as a possible approach for achieving desired substantive ends. We include this analysis in the following two appendices, respectively.

## D.1    Privacy

Given the breadth of data generative-AI models ingest for training, many experts worry models may reveal private information that they were trained on through their generations [e.g., 9, 12, 32, 109, 129]. These concerns relate to privacy rights in different jurisdictions and associated remedies to preserve those rights. As discussed in Section 2, in a number of jurisdictions, individuals have the right to request that organizations delete their personal data, also referred to as the "right to be forgotten," following Article 17 of the GDPR [112]. Regulators may seek remedies that require the removal of a set of data examples used for model training (Section 3.1) that they assess to have been unlawfully or improperly collected. In other cases, remedies may be more far-reaching; regulators may seek to delete a trained model in its entirety, which is often referred to as **algorithmic disgorgement** [1, 87, 148] or (often in copyright contexts) **model destruction** [26, 35, 120].

In the context of data protection and privacy of personal information, deletion requirements[1] also often demand removal of data within a certain time limit. For example, the California Consumer Privacy Act (CCPA) requires businesses to reply to data-deletion requests within 45 business days, extendable to 90 business days [14]. While deleting specific records from a traditional database or dataset in this time frame is often technically feasible, some laws recognize that, in other cases, deletion may be less feasible or require "disproportionate effort." In such circumstances, some jurisdictions may provide exceptions to deletion requirements, for example, in cases where the information is otherwise publicly available, is necessary to complete a transaction, is necessary to achieve purposes in the public interest, or is necessary comply with other legal obligations [e.g., Article 17(3), 112]. Some jurisdictions have also ruled that information should be suppressed from being presented to users, even if the underlying data could not be deleted [42].

Such requirements and possible remedies have motivated attention to methods in machine unlearning as an approach for achieving compliance with privacy legislation. For example, at least in principle, unlearning perhaps seems like a direct match for satisfying data deletion requests in a more efficient and targeted way: unlearning methods could be a finer-grained alternative to complete model disgorgement, less expensive and more efficient than retraining models from scratch on new datasets, and, due to improved efficiency, more suitable for satisfying deletion timeframes generally required by privacy and data protection laws. More generally, unlearning methods have appeal because they seem to strike a balance between desires to enable large-scale training of AI models and to retain a toolkit of interventions that advance privacy. But of course, as with assessing any potential

---

[1]Other legislation, e.g., the Virginia Consumer Data Protection Act [142], also has requirements for data correction, not just data deletion. Correction could be operationalized as removal of the incorrect data and replacing them with (i.e., retraining with) the corrected data.

remedy, it would likely be necessary to consider the feasibility or reasonableness of using a particular unlearning method in practice, with respect to desired targets, costs, and overall effectiveness.

We address some of these considerations below, organized around three broad goals that we observe for unlearning-related efforts for generative AI that pertain to privacy concerns grounded in regulatory frameworks: (1) data deletion (i.e., removing specific information from a model's training dataset), as well as suppression at generation time of (2) outputs that resemble personal information and (3) latent information learned from the training data.

**Data deletion requests (i.e., removal of specific training data).** Data deletion involves entirely removing specific training data from training datasets. It is often motivated by (1) legal rights of individuals (often called **data subjects**) to request the deletion of personal data coupled with (2) implicit expectations that removing training data will help to mitigate against models outputting verbatim pieces of potentially private training data.

In some cases, for generative AI, data deletion is most straightforwardly implemented by retraining a model from scratch or, if applicable or feasible,[2] some structural removal method (Section 3.1) with the right-exercising data subject's examples removed from the training dataset.[3] Depending on the reason for deletion, this might on its own suffice for certain deletion requests. For example, if the main issue being remedied is lack of consent for the use of a data subject's personal data, output suppression might not be a relevant remedy. It also may not be an issue if inferences can still be made about the data subject, so long as those inferences are based on training and prompt data that have been processed with proper consent.

However, even though removal methods seem like a direct match for implementing data deletion, they are not a straightforward solution in practice. In general, there are significant technical challenges in identifying all instances of training data that meet certain privacy-relevant criteria within large-scale datasets.[4] Consider a deletion request to remove images of a particular data subject from the training data used to produce an image generation model. This may be computationally expensive at scale and require the use of ML tools that are themselves imperfect at identifying all such examples. Even if there existed tools that guaranteed perfect identification of a given set of examples, in privacy contexts, there are fundamental challenges for drawing boundaries around what this set ought to include (Mismatch 2). Should the set to remove be conservative and include images that only feature the right-exercising data subject? Should it be more (and perhaps overly) inclusive, and also cover family photos where other data subjects are also present? Photos where the right-exercising data subject is in the background?[5]

Further, even if perfect removal of specific training data were to satisfy deletion requests in name, this would not guarantee that a model could not produce outputs that reflect these data, which could matter in contexts where model outputs are also of concern. Even in the best-case scenario, an unlearning method that satisfies privacy requirements with respect to how models are trained would be insufficient to guarantee privacy is preserved at generation time (Mismatches 2 & 3). In general, on their own, methods for removal are insufficient to guarantee privacy is preserved at generation time.

**Output suppression of training data (and information that resembles it).** Stakeholders may also focus efforts on suppressing certain pieces of training data from outputs, either through modifying the generative-AI model (e.g., with RLHF) or system-level filters (Section 3.2). This would cover

---

[2]Recall that structural removal methods are not currently widely usable for generative-AI contexts. Methods that approximate structural removal do not guarantee that the targeted training data are actually removed. (See Section 3.1 and Section 4, Mismatch 1.)

[3]Separately, some argue that models trained with methods like **differential privacy** are sufficient to preserve a data subject's privacy; in such cases, some believe that the data subject's privacy is retained even though their examples are included in the training data, so it would not be necessary to use machine unlearning methods to remove them. See Brown et al. [12] for additional discussion.

[4]Current legislative deletion provisions tend to have concrete scopes for deletion criteria, such as deletion of data associated with a particular user account [e.g., 14]. Such boundaries are less clear for training datasets, for which the underlying training examples tend not to be organized in relation to their provenance [86], e.g., according to the user to which the data relate.

[5]Methods that approximate structural removal (Section 3.1) inherit these challenges. Unlike structural removal methods, they do not guarantee with certainty that the chosen set of examples is removed from the model (Mismatch 1). The extent to which more efficient, but less accurate, approximate structural methods could be sufficient to stand in for structural ones is a question for policymakers and regulators [28].

cases in which a particular piece of training data was for some reason not included in a deletion request (e.g., because there was a failure to identify or deem it appropriate for deletion), as well as cases in which latent information in the trained model enables the generation of outputs that resemble the right-exercising individual's personal information.

If effective, output suppression would prevent surfacing these training data to end users, but would not actively remove it from trained models or their training datasets. This approach most closely resembles the notion of the "right to be forgotten" that follows from the Court of Justice of the European Union (CJEU) ruling in 2019. The CJEU ruled that Google should act on requests from data subjects by suppressing information from a viewable index in relevant jurisdictions, but did not necessarily require deleting that information from underlying data storage [42].[6] Approaches that support suppression of certain types of outputs are imperfect (Section 3.2); it would be likely that efforts to suppress targeted training data would be subject to a test of reasonable or proportionate effort, with effectiveness determined by an evaluation of how difficult it would be to extract the suppressed information from the model following the application of technical or procedural interventions, for example, through red teaming and related procedures.[7]

**Output suppression of latent information.** Additionally, many privacy practitioners have come to recognize that simply restricting the collection or processing of certain training data may not mitigate privacy concerns. This can be the case if technology enables an actor to infer information about a particular data subject based on latent information derived from similar data subjects who have consented to (or not objected to) data processing (e.g., inferring $p_0$'s health status in Mismatch 3), or to infer sensitive characteristics from benign data that may be subject to fewer restrictions. For instance, California privacy regulators view such inferred information to still be personal information about consumers over which they can exercise their rights, when such information is used to make a profile about them [14].

Stakeholders may be concerned about two distinct elements of inferences like these that are derived from latent information. First, if a generative-AI model or system has explicitly generated and then stored inferred information about an individual, such that a new data point has been explicitly created (e.g., a data point about $p_0$'s inferred health status), deleting that new data point from storage may be expected in order to meaningfully preserve that individual's privacy. However, if a model has the *capability* to draw connections about a data subject via latent information in its parameters and additional information provided in the user's prompt, output suppression approaches may be more appropriate to prevent generations that compromise that data subject's privacy.[8]

Importantly, all three areas discussed above are neither mutually exclusive nor independent. They could each be implemented in the service of satisfying privacy aims. But this is also not straightforward, as sometimes different privacy goals and the relevant technical approaches to attempt to accomplish them may be in tension. For instance, implementing an output suppression intervention may require a system operator to retain the information that should be prevented from being surfaced, in order to filter out this information from an output or filter out prompts that aim to solicit it.[9] Removal of training data, meanwhile, may create the (as we have seen, false) impression that a model will not be able to output a specific piece of information. Failing to implement efforts

---

[6]In brief, the court found that the request in question fell under the law [42, paragraph 52]; however, removal of the information from all domains (not just those that reflect the Member States of the European Union) was an over-broad interpretation of the authority and scope for the relevant laws [42, paragraphs 59-65]. The court found that it would suffice to de-reference the information "on the versions of that search engine corresponding to all the Member States, using, where necessary, measures which, while meeting the legal requirements, effectively prevent or, at the very least, seriously discourage an internet user conducting a search from one of the Member States on the basis of a data subject's name from gaining access, via the list of results displayed following that search, to the links which are the subject of that request" [42, paragraph 74].

[7]For discussions of red teaming, we refer to Feffer et al. [49] and Chouldechova et al. [21].

[8]As in Appendix C, the factuality of a piece of latent information is not relevant for our purposes. For example, making an incorrect inference about $p_0$'s health status may still violate $p_0$'s privacy. More generally, such false or "hallucinated" outputs can still cause harm.

[9]This tension—of needing to retain information in order to facilitate suppression—is also relevant for copyright (Section 5.1) and safety (Appendix D.2). More generally, this tension pre-dates interest in machine unlearning for generative AI. For instance, in the past, Facebook attempted to address the spread of NCII on its social media platform by requesting users upload the images in question to another Facebook-hosted tool, so that Facebook could identify and remove the images from the platform [69].

to prevent the generation of that information may lead to similar concerning impacts of the collection or retention of that data. Lastly, just as it is challenging to draw clear boundaries around which data to remove to satisfy a deletion request, it is similarly a difficult and open-ended problem to draw boundaries around what to suppress from model and system outputs.

## D.2 Safety

Next, we address concerns about AI safety, which span a wide range of issues and communities [e.g., 3, 4, 9, 13, 15, 27, 30, 47, 68, 114, 133, 138, 139, 146, 149, 155]. Among this variety, there is one recurring theme that is especially important to address in relation to machine unlearning: the concern that "dual-use," large-scale generative-AI models exhibit

> high levels of performance at tasks that pose a serious risk to security, national economic security, national public health or safety, or any combination of those matters, such as by ... substantially lowering the barrier of entry for non-experts to design, synthesize, acquire, or use chemical, biological, radiological, or nuclear (CBRN) weapons [138].

We draw the quote above from a past U.S. Executive Order on the Safe, Secure, and Trustworthy Development and Use of Artificial Intelligence; however, similar concerns and language can be found in a variety of legislative and policy documents, including the E.U. AI Act [47, Recital 110],[10] the International Scientific Report on the Safety of Advanced AI produced by the AI Seoul Summit [9, Chapter 4], and OpenAI's Preparedness Framework [114]. Generative-AI models and systems "are sometimes called 'dual-use' because of their potential for both benefit and harm" [140].

Some researchers and policymakers claim that, to limit potential harmful uses, machine unlearning methods could be used to remove "unsafe," "hazardous," or otherwise "undesirable behaviors" from generative-AI models [e.g., 9, 91, 92, 95, 98, 157]. For one notable example, in the cross-stakeholder AI Seoul Summit report, Bengio et al. claim that "'Machine unlearning' can help to remove certain undesirable capabilities," e.g., those "that could aid malicious users in making explosives, bioweapons, chemical weapons, and cyberattacks" [9, p. 75].

**Unclear boundaries for removal.** For now, we set aside questions of output suppression, and note that there are particular challenges for safety contexts with respect to drawing lines around what to target for removal from a model (Mismatch 2). For example, some proponents of unlearning as an approach for improving safety assume that specific topics with dual-use potential, such as synthetic biology or chemistry, can be successfully targeted for removal. But topics like these are broad and under-specified, and relate to all sorts of information both directly in the training data and latent across training data. How should one go about determining reasonable boundaries for which training examples should be kept and which should be targeted for removal?[11]

In certain cases, it may be possible to remove training data from datasets, which are intrinsically harmful or have the high potential to be put to harmful uses—for example, respectively, data containing non-consensual intimate imagery (NCII) [111] or the molecular structure of the smallpox virus.[12] However, many types of training data do not fit into these categories. Instead, potential safety issues come about from latent information: the fact that many potentially dangerous items can be assembled using training data that are themselves innocuous or have significant legitimate uses. For instance, from all of the information in a high school chemistry curriculum, it is possible to derive formulas for toxic molecules. But removal of all knowledge of high school chemistry to foreclose the possibility of the model containing or producing such latent information is likely overbroad.

So far, we have only considered the trained model; safety challenges are also difficult when we consider the model's potential generations. As discussed in Section 2 and with respect to Mismatch 3,

---

[10]"... international approaches have so far identified the need to pay attention to risks from potential intentional misuse or unintended issues of control relating to alignment with human intent; chemical, biological, radiological, and nuclear risks, such as the ways in which barriers to entry can be lowered ..." [47, Recital 110].

[11]For instance, even after unlearning information related to biohazards to reduce unsafe question-answering capabilities, researchers have shown it is possible to recover such capabilities by further training the model on unrelated benign information. Distinguishing which exact training data (regardless of whether it is considered "safe" or "unsafe") contribute to these unsafe capabilities is unclear [160].

[12]Of course, a formula for a molecular structure is not sufficient to produce a molecule (Mismatch 4); but for sufficiently dangerous molecules, the formula itself might be considered a safety risk.

the open-ended format of user inputs to generative-AI models means that, via their prompts, end-users can introduce additional information into the model's context at generation time. This information could be otherwise absent from the model's training data and not reflected in the model's parameters. It could also overlap with or reflect training data that was removed from the model using an unlearning method. By bringing this information back into the generative-AI system via the prompt, the model can still be used to reason about the information it has unlearned; its output might even be the same as if an unlearning method had not been applied in the first place (Section 4, Shumailov et al. [124]).[13]

**Inherent tensions for unlearning in dual-use systems.** Separate from the practical difficulties discussed above, there is an even more fundamental challenge originating from the inherent nature of dual-use systems. By definition, dual-use systems can be put to potentially beneficial or potentially harmful uses [140]. It is not just the case that innocuous training data could in combination lead to potentially unsafe latent information in the trained model; it is also possible for generated outputs that are innocuous in isolation to be put to unsafe or otherwise undesirable downstream uses [58].

Consider the example of unlearning all information for "how to synthesize a toxic molecule." Setting aside the tractability of translating this into concrete targets to unlearn, knowledge of how to actually produce such a molecule is not a property of the model in isolation. The ability to actually synthesize it also depends on the knowledge of the user [128].[14] The particular user is clearly an important factor to consider with respect to downstream use. What if the user already has a recipe for making such a molecule (obtained from another source), and the generative-AI model lowers the barrier for creation of the molecule for the user by explaining, in detail, how to understand nuances of the recipe that they do not understand on their own? What if the model provides a single "missing piece" of information that is innocuous on its own (e.g., details of a single chemical reaction) that, in combination with everything else this user knows, enables them to create the molecule?[15]

For an additional example, consider a generative-AI system that includes a model trained for molecular generation—for suggesting formulas for new drugs and other molecules. Arguably, one of the purposes of such a system is to lower the barrier of expertise required for drug discovery. Even so, currently, a generative-AI system cannot on its own definitively determine that the molecules it produces are safe for human consumption; this is the point of lab experiments and drug trials [e.g., 136]. Once again, safety in this case is not an isolated property of the generative-AI model or system. Additional knowledge—in this case, derived through biochemical experimentation and human trials—is often needed to determine if the generative-AI outputs are beneficial or harmful.

As discussed with respect to Mismatch 4, the types of control that methods for machine unlearning provide are incapable of preventing such harmful downstream outcomes. Unlearning can perhaps limit the information in the model or suppress model outputs, such that certain types of unsafe outputs are less likely. But unlearning cannot guarantee that people or other agents will not put model outputs to unsafe uses [76].

---

[13]Evaluations for the success of unlearning methods in safety contexts often do not explicitly test for this issue (Section 3.2). Many such evaluations rely on the WMDP benchmark [91], which is a multiple choice question dataset that focuses on biological, chemical, and cybersecurity risks. Setting aside the observation that multiple choice questions may not in general be the most effective way to measure such risks, this evaluation setup does not allow for the type of more open-ended reasoning that this scenario presents.

[14]It is perhaps for this reason that companies' safety frameworks often categorize CBRN risks in relation to both the model and the users [54, 103]. For example, OpenAI's Preparedness Framework considers high risk to mean that the "Model enables an expert to develop a novel threat vector OR model provides meaningfully improved assistance that enables anyone with basic training in a relevant field (e.g., introductory undergraduate biology course) to be able to create a CBRN threat" [114, p. 9].

[15]One might critique this example for not qualifying to "substantially lower[] the barrier of entry for non-experts" to perform unsafe actions in the world. Perhaps the user could have found similar information through effective use of a non-generative-AI system, like a traditional search engine indexed over the public Internet, so it is questionable that using a generative-AI system was sufficiently "substantial" to make the task easier to execute [79]. We set this question aside. It is not in scope for us to make claims about whether or not existing generative-AI models and systems meet the bar of what, for example, the now-rescinded U.S. executive order considered a meaningful safety risk [138]. Regardless, we can still address the extent to which unlearning can or cannot address cases like this one.

