# OpenReview forum: "Machine Unlearning Doesn't Do What You Think:  Lessons for Generative AI Policy and Research"
_NeurIPS.cc/2025/Position_Paper_Track — NeurIPS 2025 Position Paper Track Oral_

### Official Review · Reviewer_C3YD · 2025-07-15

**Significance:** 2
**Presentation:** 3
**Rating:** 4
**Confidence:** 3

**Summary:**

This paper proposes a position that the understanding of machine unlearning should be rectified in terms of the generative AI policy. The paper first revisits the two goals of machine unlearning: removal and suppression, and highlight they are very hard to achieve in general cases. Then the paper identifies several "mismatches" between the goal and the feasible implementations. The paper also studies this issue in US Copyright and finally gives some insights for ML research and AI policy.

**Strengths:**

This paper is clear and easy to follow. The discussion is accessible to broad readers that are not familiar with machine unlearning or even machine learning. This paper is based on strong case analysis, and deep understanding of generative AI policy. It's timely in the era of generative AI. I think it's quite insightful for decision making for government.

**Weaknesses:**

I have several concerns:

1. This paper may not catch many eyes in NeurIPS community, though it will be beneficial for AI policy makers. In my understanding, ML community cares more about technical issues for topics like machine unlearning, but this paper focuses more on the application-wise issues. I am not sure if ML community are willing to deliver a solid "application" or "system" for target information removal and suppression in real applications in real world, but I think this must be a minor focus among ML community.
2. I am not very convinced by the discussion of mismatches. The feasible implementation does not exist for the time being doesn't necessarily means the goal cannot be achieved. Instead, it shows there is much room for improvement.
3. To me (personally as a ML practitioner), it seems to lack deep or strong technical insights. The part of "Takeaways for ML research" doesn't bring me excitement, and I have an impression that those claims are just correct and easy to accept, but not impressive.
4. I think the paper should gives more references on misunderstanding of machine unlearning. Without such evidence, the critique risks addressing an unsubstantiated or "strawman" position.

**Questions:**

I don't have questions.

**Alternative Position:**

No

**Author Identification:**

No.

**Context:**

2

**Discussion:**

2

**Ethics:**

["NO or VERY MINOR ethics concerns only"]

**Position:**

Yes, the paper argues for or against a position related to machine learning.

**Support:**

2

**Thoroughness:**

3

---

### Official Review · Reviewer_ARK4 · 2025-07-20

**Significance:** 4
**Presentation:** 3
**Rating:** 7
**Confidence:** 3

**Summary:**

The position paper discusses the complexities of machine unlearning presents an argument that there is a disconnect or mismatch between how unlearning works against the way it is claimed by people to be one of the solutions in addressing issues such as copyright infringement (as well as safety, privacy, etc) in modern generative AI models. The discussion centers around two aspects of unlearning namely targeted suppression and removal of information from the model. The authors emphasize that these two concepts are not interchangeable which may further confuse if propagated as a form of compliance with ongoing legislation related to removing/suppressing/filtering copyrighted content from AI models. To strengthen the support for the main argument, the authors structure the paper by discussing evolving motivations of machine unlearning, five main conceptual mismatches in the motivation of using unlearning against how it actually works, and further cover how said mismatches complicates its application using the US copyright law as a use case. The authors end the discussion with a set of takeaways targeted for both technical and policy people on how to move forward with machine unlearning as a form of method for compliance to policies.

**Strengths:**

The position paper has several key strengths that I appreciate:

First, I think overall topic and the authors position is an appropriate venue for the position paper track. Although I don’t work on unlearning, I’m particularly aware of the noise this field makes across discussions in social media platforms whether it actually works or not (to remove specific information or content from learning models). I believe this paper will introduce a more realistic viewpoint to the discussion in the ML community (if it has not already).

The paper presents a strong combination of perspectives on the technical and policy implications of machine unlearning gives a more balanced discussion on the authors’ position with unlearning. Given that government agencies around the world are rushing for drafting regulations and policies on protecting data privacy and intellectual property, I have a feeling this paper might be influential in their decisions, hence a solid reason to reserve a slot to be presented in conference.

To some extent, the paper is informative for people with basic knowledge of how AI works. You also don’t need to be an expert in policy to understand the nuances as these are all presented clearly in the paper.

**Weaknesses:**

The position of the paper could have been made more clear cut and direct (e.g., stating “Machine unlearning is inherently an insufficient method for meeting legal requirements in data protection and in removing data from an AI  model” or something more appropriate) rather than taking a more implicit angle from what I’m picking by how the authors discuss the arguments. Some readers might misinterpret the current challenges and uncertainties in the legal aspect discussed in the paper (e.g., copyright infringement) as something that confounds the direction of machine unlearning research, hence might say the position of the paper is also unclear/uncertain/highly dependent on legal interpretation. Thus, I think there is a need for a stronger articulation of the position.

I would like to see better framing of alternative views discussing where machine learning may potentially serve as the sole or main solution to certain problems (e.g., forcing models to unlearn illegal content such as sexual content involving a minor). While I do get the importance of emphasizing and supporting the position, a more balanced discussion on the alternative views is needed for the paper.

**Questions:**

While unlearning research will continue to progress, we are doubtful that full compliance will be achieved in the future. → Is this an assumption that unlearning should always be used in complement or on top of other methods for removing/analyzing presence of targeted content? I’m quite confused by this statement.

In what specific edge cases is machine unlearning a feasible solution to be used alone? Or is this applicable only if it’s not towards compliance to legal requirements?

**Alternative Position:**

Yes, and alternative positions are well-considered and addressed by the argument

**Author Identification:**

No.

**Context:**

4

**Discussion:**

4

**Ethics:**

["NO or VERY MINOR ethics concerns only"]

**Position:**

Yes, the paper argues for or against a position related to machine learning.

**Support:**

3

**Thoroughness:**

4

---

### Official Review · Reviewer_79Kc · 2025-08-07

**Significance:** 3
**Presentation:** 4
**Rating:** 7
**Confidence:** 4

**Summary:**

The paper is a wake up call about control of information in the AI era, and how society at large (mainly policymakers and courts) has to manage it.

The position of the paper is that UNlearning in AI models does not work as policymakers expect, and that the term UNlearning is measleading at this stage of AI development.

Authors describe why unlearning is needed (Sec. 2), how it is currently implemented (Sec. 3), current limitations of such implementations (Section 4), impact on US Copyright law (as an example, Sec. 5). The paper ends with Sec. 6: recommendations on how policymakers should adjust their expectations.

The main techniques for AI unlearning are "targeted removal" and "targeted suppression". Authors address both of them in different scenarios and stress 5 "unsolvable" mistmatches between unlearning Motivations and Methods (listed in Sec. 4).

Their example on US-copyright law shows that there are no general-purpose solutions to constrain generative AI models and that the certainty of erasure/removal is unattainable, as AI models are inherently probabilistic. Policymakers should be warned of limitations (and costs) of using unlearning methods for compliance.

**Strengths:**

The paper raises a relevant issue in AI: how can we remove information from an AI model.

The analysis is detailed and clear on the technical and practical aspects barring a full compliance  to a removal order.

It is definitely targeting the attention of policymakers, lawyers, and judges, as it is educating them on the randomness of AI. This is a huge problem in itself, as expectations among that set of professionals are more clear cut, they have to answer questions such as "is a given event a copyright infringement or not?".

**Weaknesses:**

In some sense, AI models may be interpreted as tools in the hands of users, like a knife or a gun. Following up with this approach, it would seem natural to discuss how can we elaborate a set of social rules on how to use AI in a legal setting. Should we have a "certification" process for AI models, with warning labels?
Even though, this is not the core of the position paper, I think this part of the discussion is missing.

**Questions:**

I have a provocative one: why do policymakers have such great expectations on the unlearning capacity of AI models?
Should we ask the AI model whether the accurrence is indeed copyright violation?

**Alternative Position:**

Yes, and alternative positions are well-considered and addressed by the argument

**Author Identification:**

No.

**Context:**

4

**Discussion:**

3

**Ethics:**

["NO or VERY MINOR ethics concerns only"]

**Position:**

Yes, the paper argues for or against a position related to machine learning.

**Support:**

3

**Thoroughness:**

3

---

### Note · Authors · 2025-08-28

**1-11 Submit Again:**

Probably yes

**1-1 Submission Process:**

3

**1-2 Next Year:**

We are really excited to see NeurIPS add a position track this year. In our opinion, it is a really welcome addition to the flagship AI/ML conference.

It is possible that we missed something in the original CfP or in an email, but all of the authors on our paper were surprised that there was no discussion period. We would have liked to have had the opportunity to engage with reviewers about their comments.

**1-3 Future Development:**

We hope this track becomes part of the usual program at NeurIPS. As noted in our first answer, we think that the track could be improved with a discussion period between the authors and the reviewers, similar to the other NeurIPS tracks. We would also hope that there are other ways reviewers can be made aware of clear instructions about how this track is different from the others. We found that many review suggestions and comments (including positive ones) were not appropriately scoped to this track. We hope in the future more review suggestions and criticisms would be scoped to what is appropriate for this track’s CfP.

**1-4 Interest:**

["Panel discussions with other position paper authors", "Structured debates on controversial topics"]

**1-5 Thoughtful:**

4

**1-6 Supportive:**

6

**1-7 Technical Aspects Versus Position:**

6

**1-8 Gate Keeping:**

3

**1-9 Camera Ready Changes:**

Based on the reviewers’ feedback and our own list of intended updates, we will make the following changes:

1. We will add additional references and discussion of unlearning papers that have come out since the NeurIPS submission deadline, focusing on papers that frame themselves as a solution for copyright compliance.

2. At the end of the paper (takeaways section), we will discuss some potential avenues for future and/or related work (e.g., investigating a “certification” process for models). This will dovetail with our existing discussion about “reasonable best efforts” around unlearning, i.e., how such a “certification” process could be useful for determining whether this bar has been met.

3. We will split out a clear heading in the takeaways section or conclusion that better sign-posts our discussion of alternate views.

**3-1 Review Response1:**

79Kc

**3-2 Reaction To Review1:**

We appreciate the reviewer’s recognition of our paper’s timeliness and clarity, as well as their positive evaluation of its quality.

1.  Certification

We agree there is more to be said about what it means to “certify” AI models. As the reviewer notes, however, this question is beyond the scope of the present paper. In our takeaways, we emphasize the importance of shifting the policy discussion toward remediation, i.e., whether a developer took reasonable steps, rather than requiring perfect results. We will revise the paper to connect this to potential future work on how such “reasonable best efforts” might be certified.

2. Social rules about the use of AI in legal settings

We appreciate this comment as an interesting research direction, but addressing it here would extend beyond the intended scope of our paper and beyond our current team’s expertise in ML, law, and policy. A rigorous treatment of social norms and rules would require engagement with additional disciplinary expertise (e.g., sociology, philosophy). We see this as a rich topic for a separate paper.

3. Question about asking the AI about copyright infringement

Even as models’ knowledge of statutes and caselaw improves, there is ample evidence--such as numerous recent cases involving lawyers using chatbots for briefs and motions--that models can generate embarrassing (and sometimes grave) hallucinations in high-stakes legal settings. We therefore do not encourage asking AI models to provide legal judgments beyond casual use.

4. Question about why policymakers put so much stock in unlearning

We appreciate the reviewer’s question regarding policymakers’ credulousness and believe our paper makes a significant contribution in answering it by (i) reviewing the literature and showing outsized claims of the efficacy of unlearning for policy-related compliance issues and (ii) demonstrating that conceptual mismatches lead policymakers and technical researchers alike astray.

**3-3 Review Response2:**

ARK4

**3-4 Reaction To Review2:**

We appreciate that the reviewer recognizes both the timeliness of our topic and thinks highly of our paper.

1.  Reviewer’s suggested restatement of our position

Respectfully, a blanket statement that unlearning is inherently insufficient for legal compliance is too broad and inconsistent with our analysis. Our paper attends carefully to the specifics: different formulations of machine unlearning (removal, suppression, etc.), limitations of current methods, and technical complexities of copyright law and policy. Navigating these specifics and the tensions at their intersections is essential for developing sound policy about unlearning as a remedy, and for more grounded claims in ML research about what unlearning methods can achieve.

2. Interpretation of uncertainty in our paper

Lawyers often joke that the answer to any legal question is “it depends.” This reflects the fact that legal interpretation and deliberation are inherently complex and context-specific. A key aspect of our position is to show why overbroad statements risk misrepresenting both the technology and the law. We don't want to fall into the same trap by implying certainty where it doesn't exist. Determinations about the appropriateness of unlearning in particular contexts will rest with experts, magistrates, and regulators. We provide guidance for how to reason about this soundly, but believe it would be irresponsible to offer a sense of certainty about outcomes.

3. Alternative views

We appreciate the suggestion to draw out alternative perspectives more clearly and will add a bolded paragraph. (We address alternative views throughout our careful analysis of the ML literature.) However, we think the review’s proposed alternative view makes a claim our paper cannot responsibly include: asserting that unlearning should serve as the “sole or main solution” to certain problems. Even in severe cases like CSAM, unlearning might not be the right remedy; model destruction may be the only lawful option.

**3-5 Review Response3:**

C3YD

**3-6 Reaction To Review3:**

We surface this review in Section 2. In our view, this review mischaracterizes key aspects of our paper, applies main-track criteria, and doesn’t engage with the paper, the CfP, or reviewer guidelines.

1. Categorizing our paper as “untechnical”

Position papers are not expected to introduce new technical methods. Our position is directly relevant to technical issues because researchers often motivate papers with topics like copyright compliance; the claims in those papers can make it sound like proposed methods are working solutions, when the reality is much more complicated. Our paper aims to help guide the future of the field: if ML research continues to rely on policy motivations, it should ask scientific questions that are better equipped to rigorously address them.

2. Of “minor interest” to the ML community

Unlearning papers are often motivated by policy issues; our position speaks directly to those motivations. Our analysis is timely: since submission, arXiv has dozens of new papers that cite copyright compliance as a motivation for unlearning.

3. Insufficient references to support argument

We cite dozens of unlearning papers (e.g., Eldan & Russinovich “Who’s Harry Potter? Approximate Unlearning in LLMs”), AI policy work, and popular press articles that present unlearning as a solution to policy challenges. We synthesize these claims, rather than critiquing individual papers, and situate them in the broader ML research context. We try to provide a positive set of interventions to help direct the field in what we believe are more sound directions, rather than single out individual authors for endorsing common misunderstandings.

4. Mischaracterizing “mismatches” section

The review states conflates first-principles *conceptual* issues we identify with *feasibility* issues. We discuss feasibility in our sections on concrete methods and takeaways. If the PC wishes for us to make feasibility more prominent, we can do so, e.g., with revised signposting.

---

### Meta-Review · Area_Chair_Sk2S · 2025-09-14

**Rating:** 6
**Confidence:** 5

**Strengths:**

The paper focuses on an important topic, highlighting the critical issue of AI, and aiming to raise awareness especially among policymakers.

**Weaknesses:**

Some reviewers raised concerns about the paper’s limited technical depth and lack of appeal to the research community, noting that it seems more tailored for policymakers than researchers and might be better suited for other venues. While I agree with these concerns, I also believe the paper addresses an important topic that could influence the future direction of machine unlearning.

**Questions:**

N/A

**Ethics:**

NO or VERY MINOR ethics concerns only

**Thoroughness:**

5

---

### Decision · Program_Chairs · 2025-09-26

Accept (Oral)